# An Effective Sanitizer for Fresh Produce Production: *In Situ* Plasma-Activated Water Treatment Inactivates Pathogenic Bacteria and Maintains the Quality of Cucurbit Fruit

Joanna G. Rothwell,[a] Jungmi Hong,[b] Stuart J. Morrison,[c] Heema Kumari Nilesh Vyas,[b,d] Binbin Xia,[b] Anne Mai-Prochnow,[b] Robyn McConchie,[a] Kim-Yen Phan-Thien,[a] Patrick J. Cullen,[b] Dee A. Carter[a,d]

[a]ARC Training Centre for Food Safety in the Fresh Produce Industry, School of Life and Environmental Sciences, Faculty of Science, Sydney Institute of Agriculture, The University of Sydney, Sydney, New South Wales, Australia
[b]School of Chemical and Biomolecular Engineering, The University of Sydney, Sydney, New South Wales, Australia
[c]Department of Agricultural and Resource Economics, University of California, Davis, California, USA
[d]Sydney Institute of Infectious Diseases, The University of Sydney, Sydney, New South Wales, Australia

**ABSTRACT** The effect of plasma-activated water (PAW) generated with a dielectric barrier discharge diffusor (DBDD) system on microbial load and organoleptic quality of cucamelons was investigated and compared to the established sanitizer, sodium hypochlorite (NaOCl). Pathogenic serotypes of *Escherichia coli*, *Salmonella enterica*, and *Listeria monocytogenes* were inoculated onto the surface of cucamelons (6.5 log CFU g$^{-1}$) and into the wash water (6 log CFU mL$^{-1}$). PAW treatment involved 2 min *in situ* with water activated at 1,500 Hz and 120 V and air as the feed gas; NaOCl treatment was a wash with 100 ppm total chlorine; control treatment was a wash with tap water. PAW treatment produced a 3-log CFU g$^{-1}$ reduction of pathogens on the cucamelon surface without negatively impacting quality or shelf life. NaOCl treatment reduced the pathogenic bacteria on the cucamelon surface by 3 to 4 log CFU g$^{-1}$; however, this treatment also reduced fruit shelf life and quality. Both systems reduced 6-log CFU mL$^{-1}$ pathogens in the wash water to below detectable limits. The critical role of superoxide anion radical ($\cdot O_2^-$) in the antimicrobial power of DBDD-PAW was demonstrated through a Tiron scavenger assay, and chemistry modeling confirmed that $\cdot O_2^-$ generation readily occurs in DBDD-PAW generated with the employed settings. Modeling of the physical forces produced during plasma treatment showed that bacteria likely experience strong local electric fields and polarization. We hypothesize that these physical effects synergize with reactive chemical species to produce the acute antimicrobial activity seen with the *in situ* PAW system.

**IMPORTANCE** Plasma-activated water (PAW) is an emerging sanitizer in the fresh food industry, where food safety must be achieved without a thermal kill step. Here, we demonstrate PAW generated *in situ* to be a competitive sanitizer technology, providing a significant reduction of pathogenic and spoilage microorganisms while maintaining the quality and shelf life of the produce item. Our experimental results are supported by modeling of the plasma chemistry and applied physical forces, which show that the system can generate highly reactive $\cdot O_2^-$ and strong electric fields that combine to produce potent antimicrobial power. *In situ* PAW has promise in industrial applications as it requires only low power (12 W), tap water, and air. Moreover, it does not produce toxic by-products or hazardous effluent waste, making it a sustainable solution for fresh food safety.

**KEYWORDS** fresh produce, cold plasma, Cucurbitaceae, superoxide, antimicrobial treatment, food safety, *E. coli*, *Salmonella*, *Listeria*, spoilage

Address correspondence to Dee A. Carter, dee.carter@sydney.edu.au.

The authors declare a conflict of interest. P. J. Cullen is the Chief Technology Officer of Plasmaleap Technologies, the supplier of the plasma technology employed to generate plasma bubbles in this study.

Fresh fruit and vegetables are an important component of a healthy diet and are frequently eaten raw or with minimal processing. However, during production, fresh produce can potentially become contaminated with human pathogens, resulting in foodborne disease upon consumption (1). Contaminated fresh produce is an important source of foodborne diseases globally, with 988 outbreaks and 45,723 cases reported from 2010 to 2015 across New Zealand, Australia, the United States, the European Union, Canada, and Japan (2). To mitigate the risk of pathogen contamination, postharvest sanitizer treatments are widely employed in the fresh produce industry (3–5). Many types of fresh produce are treated with sanitizer washes to remove debris and reduce spoilage organisms adhered to the produce surface (6, 7). Sanitizers are also critical for reducing the risk of cross-contamination by pathogens that may have been transferred into the wash solution (8–11).

Sanitizers containing active chlorine compounds such as sodium hypochlorite (NaOCl) are widely used in the postharvest treatment of fresh produce. However, chlorine reacts with soil and other organic compounds from the fruit and vegetables in the wash water, leading to the formation of toxic chlorinated disinfection by-products (DBPs) (12). The creation of DBPs reduces the amount of free chlorine available for sanitation (13, 14), which may lead to survival and subsequent cross-contamination of fresh produce with pathogenic bacteria (11). DBPs created from food sanitization are also hazardous for workers in the processing environment and are potentially carcinogenic (15–17). For Australian fresh produce to be certified as organic, chlorine sanitizers cannot be used (6), and globally there is an increasing trend for countries to eliminate their use in fresh produce production (18). This highlights the need for alternative sanitizer technologies that are better for the environment and consumers while also being effective in maintaining the safety and quality of fresh produce.

Cold atmospheric plasma is an emerging sanitizer technology with a variety of applications including in food production (19). Cold plasma is generated by applying electrical discharges to a gas so that orbital electrons are stripped from atoms, in a process called ionization. This results in a highly reactive mixture of excited species, free electrons, ions, and photons. Plasma gas can be discharged into water, which changes the physicochemical properties of the solution and results in the generation of a large variety of reactive species, such as hydrogen peroxide ($H_2O_2$), nitrite ions ($NO_2^-$), nitrate ions ($NO_3^-$), superoxide anion radicals ($\cdot O_2^-$), and hydroxyl radicals ($\cdot OH$) (20). The generated solution, called plasma-activated water (PAW), has demonstrated antimicrobial power in the treatment of strawberries (21), blueberries (22), grapes (23), tomatoes (24), mushrooms (25), and leafy greens (26, 27). PAW technology represents a critically needed alternative to toxic chlorine-based sanitizers, and it requires only air, tap water, a plasma generator, and electricity to run.

In order to apply PAW to fresh produce decontamination, it is important to consider the economic viability of technology destined for eventual scale-up and application in industrial processing. For example, current research frequently uses purified or distilled water as the PAW substrate (28, 29). However, tap water is more reflective of current (and potential future) industry practice, even though it may produce lower antimicrobial power and less reproducible results than purified or distilled water (30). Similarly, current PAW research frequently uses discharge gases such as argon or oxygen in plasma generation. These gases are expensive and unfeasible to use in fresh produce industries, which typically operate on narrow profit margins (29). Produce shelf life and quality are important considerations for the food industry that may be influenced by PAW treatment, but these interactions have not yet been adequately assessed. Finally, PAW research needs to have compatibility with current process flows used in industry by reducing or maintaining sanitizer treatment times and by reducing or eliminating the need for additional steps such as preactivation of the water.

In our previous work, we demonstrated rapid antimicrobial power against bacterial foodborne pathogens using a dielectric barrier discharge diffusor (DBDD) PAW system with tap water as the PAW substrate and air as the discharge gas (30). Physicochemical

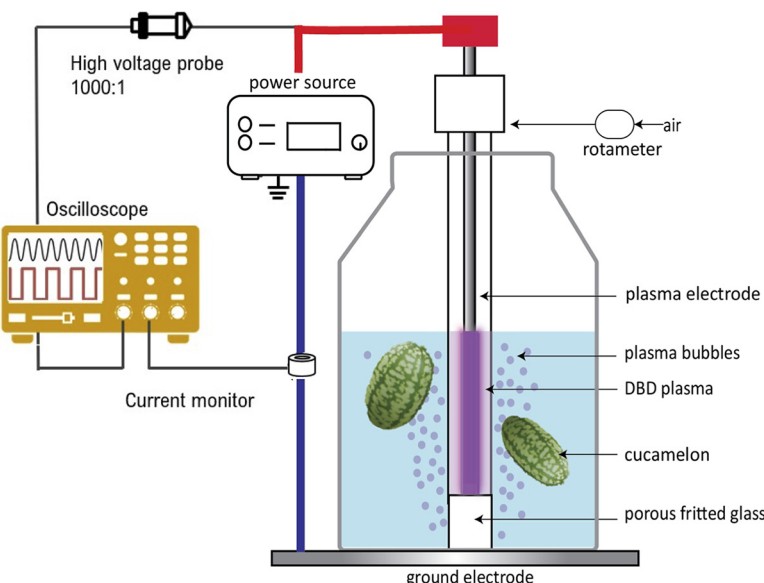

**FIG 1** Schematic of the experimental design for the treatment of cucamelons by PAW generated by a dielectric barrier discharge diffusor (DBDD) system.

analysis of the PAW revealed that a DBDD reactor using tap water produced extremely low concentrations of reactive nitrogen species ($NO_x$) and $H_2O_2$, and assays using the $\cdot O_2^-$ scavenger Tiron demonstrated that $\cdot O_2^-$ was essential for the antimicrobial activity of this system.

In the current study, we tested the efficacy of the DBDD-PAW system using cucamelons (*Melothria scabra*) as a fresh produce model (Fig. 1). These are a type of cucurbit that produce small fruit similar in flavor and texture to cucumber, with skin resembling that of a watermelon. We compared DBDD-PAW with commercially relevant concentrations of NaOCl for its capacity to reduce bacterial pathogens while preserving cucamelon shelf life. To further investigate the unique antimicrobial properties of the *in situ* DBDD-PAW system, we simulated the local electric field distribution and polarization on bacterial cells in solution and on the cucamelon surface (Fig. 2). Intense local electric fields are shown to contribute to the antimicrobial power of *in situ* PAW systems via membrane damage and electroporation (31). We hypothesize that $\cdot O_2^-$ and/or downstream reactive species, combined with the membrane damage induced by electric fields and polarization, lead to the antimicrobial activity observed in this system.

## RESULTS

**Reduction of pathogenic bacteria on the cucamelon surface and in wash water.** The reduction of pathogens on the surfaces of cucamelons and in the wash water is shown in Fig. 3. Two-minute treatments with PAW or NaOCl reduced the counts of all pathogens on the cucamelon surface by a total of 3-log CFU $g^{-1}$ and 3- to 4-log CFU $g^{-1}$, respectively, and in comparison to the water control, by 1- 1.5-log CFU $g^{-1}$ and 1.2- to 2.4-log CFU $g^{-1}$ respectively. In wash water, both sanitizers reduced 6 log CFU $mL^{-1}$ of pathogenic bacteria to below detectable limits within the 2-min treatment time. The addition of the $\cdot O_2^-$ scavenger Tiron to DBDD-PAW treatment significantly increased survival of the pathogens in the wash water and on the cucamelon surface. This demonstrated that $\cdot O_2^-$ and/or reactive species produced by $\cdot O_2^-$ in solution are required for bacterial killing by DBDD-PAW.

**SEM of pathogenic bacteria on the cucamelon surface after sanitizer treatment.** Scanning electron microscopy (SEM) was used to assess the impact of PAW and NaOCl treatment on the morphology of pathogenic microbes adhered to the surface of the cucamelons (Fig. 4). The water-treated control cells were typically intact, smooth, and plump. Both NaOCl and PAW treatment led to distinct morphological changes. PAW

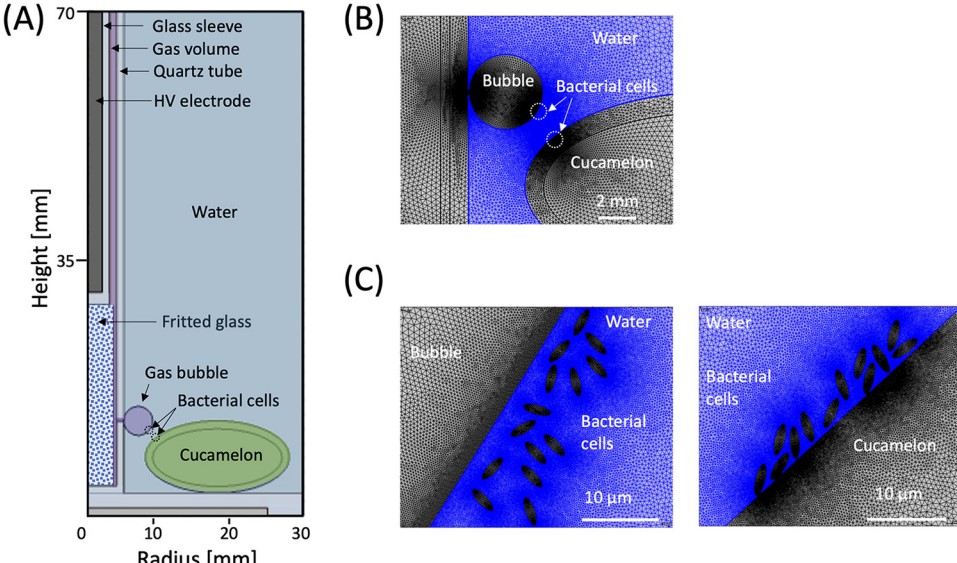

**FIG 2** Schematic of the DBDD-PAW reactor containing the cucamelon and bacteria for electric field modeling with COMSOL. (A) DDBD reactor geometry including bacterial cells attached to the cucamelon skin for electric field analysis; (B) built-in mesh geometry for the COMSOL program near the bubble and cucamelon surface; (C) details of the modeled bacterial cells floating in solution close to the bubble (left) and on the cucamelon skin (right).

treatment caused *Escherichia coli* to have a deflated and ruptured cellular morphology, while *Listeria monocytogenes* cells had holes in the cell wall at the polar ends of the rods (white arrows, Fig. 4). NaOCl treatment caused moderate crumpling or puckering of the cell surface in both species.

**Quality and organoleptic properties of treated cucamelons over storage time.** The effects of wash treatment on background microbiota are shown in Fig. 5A and B. Initial counts of total viable mesophilic bacterial counts (TVCs) on cucamelon surfaces were 5.9 log CFU g$^{-1}$, which was reduced by 0.8 log CFU g$^{-1}$ following washing with tap water, 1.7 log CFU g$^{-1}$ by PAW treatment, and 2.2 log CFU g$^{-1}$ by NaOCl treatment. By the end of the storage trial, the TVCs were similar across the different treatments. Untreated cucamelons had 4.9 log CFU g$^{-1}$ of background yeast and molds that was

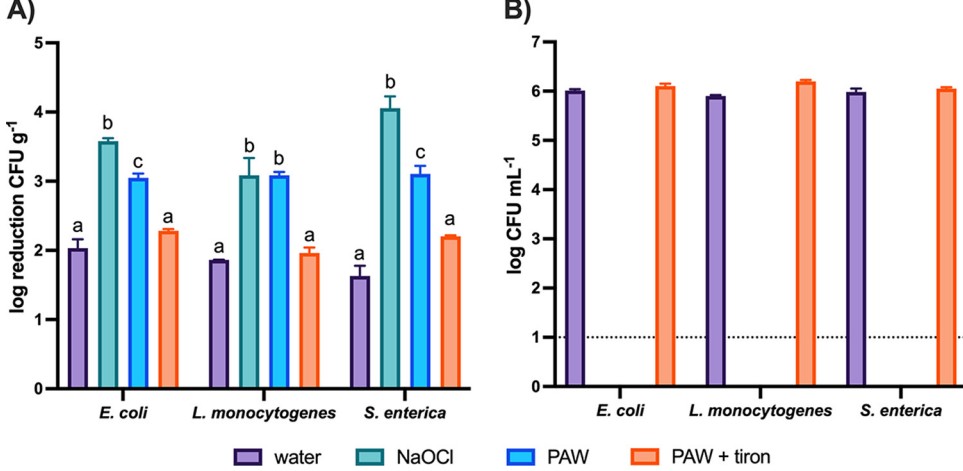

**FIG 3** Survival of pathogenic bacteria in the wash water and adhered to the cucamelon surface following a 2-min wash treatment. (A) Log reduction of bacterial CFU per gram on cucamelon surfaces compared to an unwashed control after 2-min treatment with tap water, NaOCl (total chlorine, 100 ppm, and pH 6.5), PAW, or PAW with the $\cdot O_2^-$ scavenger Tiron. (B) Survival of pathogenic bacteria in the wash solution after tap water, NaOCl, PAW, or PAW and Tiron treatments. *P* values of <0.05 are denoted by different letters. Error bars represent the standard error of the mean, and the dotted line denotes the limit of detection.

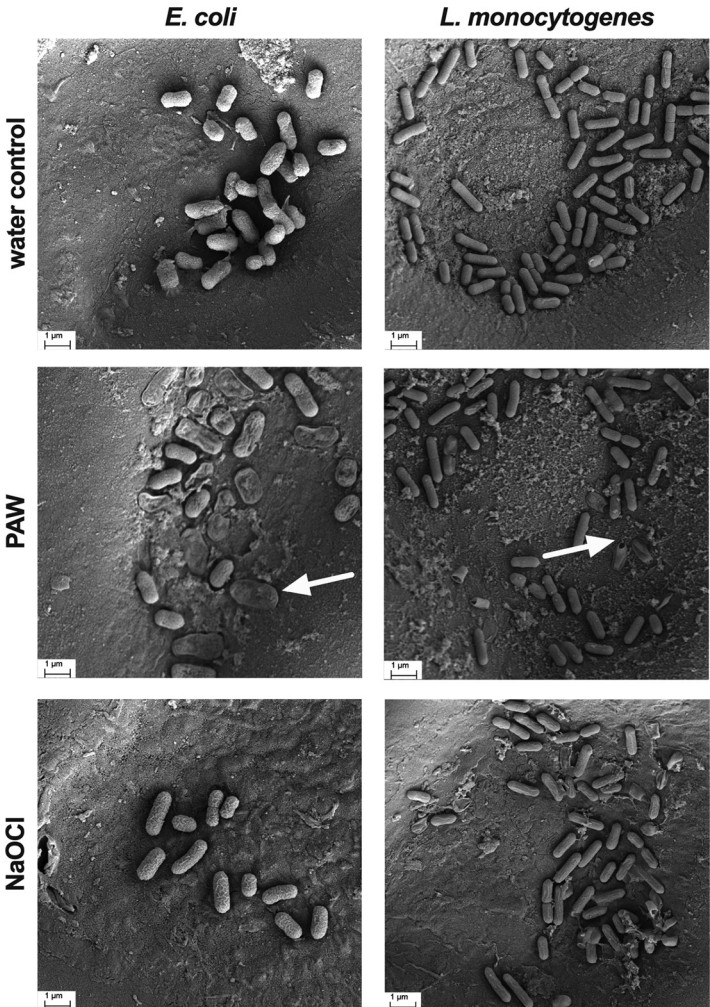

**FIG 4** Scanning electron microscopy images of pathogenic bacteria adhered to the surface of cucamelons and treated with a 2-min wash of water, PAW, or NaOCl. Following PAW treatment, many of the *E. coli* cells exhibited a deflated, dehydrated, and crumpled cellular morphology (arrow on left panel), while some *L. monocytogenes* cells had a distinct rupturing from their outermost ends (arrow on right panel). The surfaces of the cells of both bacterial species were moderately crumpled following treatment with NaOCl.

not significantly reduced by treatment with water, while NaOCl and PAW treatment reduced counts by 1.5 log CFU g$^{-1}$ and 2 log CFU g$^{-1}$, respectively. The background microbiota that washed off the cucamelons and into the wash water was 3 log CFU mL$^{-1}$ of TVCs and 2.6 log CFU mL$^{-1}$ of yeast and molds. Treatment with PAW or NaOCl reduced these microbes in the wash water to below detectable limits (data not shown).

Various quality parameters of the treated cucamelons were investigated. The firmness of the cucamelons directly after sanitizer treatment and over the 21 days of storage is shown in Fig. 5C. Cucamelons that were treated with NaOCl were the softest of all the treated fruit from day 7 onward; however, the only significant difference in texture in this series was that PAW-treated cucamelons were significantly firmer on the final day of the experiment than those treated with NaOCl. The color metrics of light, dark, and combined sections of cucamelons following different wash conditions were analyzed, with the total combined color change over time for the light and dark sections shown in Fig. 5D. The wash treatments did not result in significantly different color changes between day of treatment and day 21 when compared using pairwise *t* tests. On day 28, the sensory quality scores of the cucamelons treated with PAW remained above 3, while those for the NaOCl-treated cucamelons were significantly

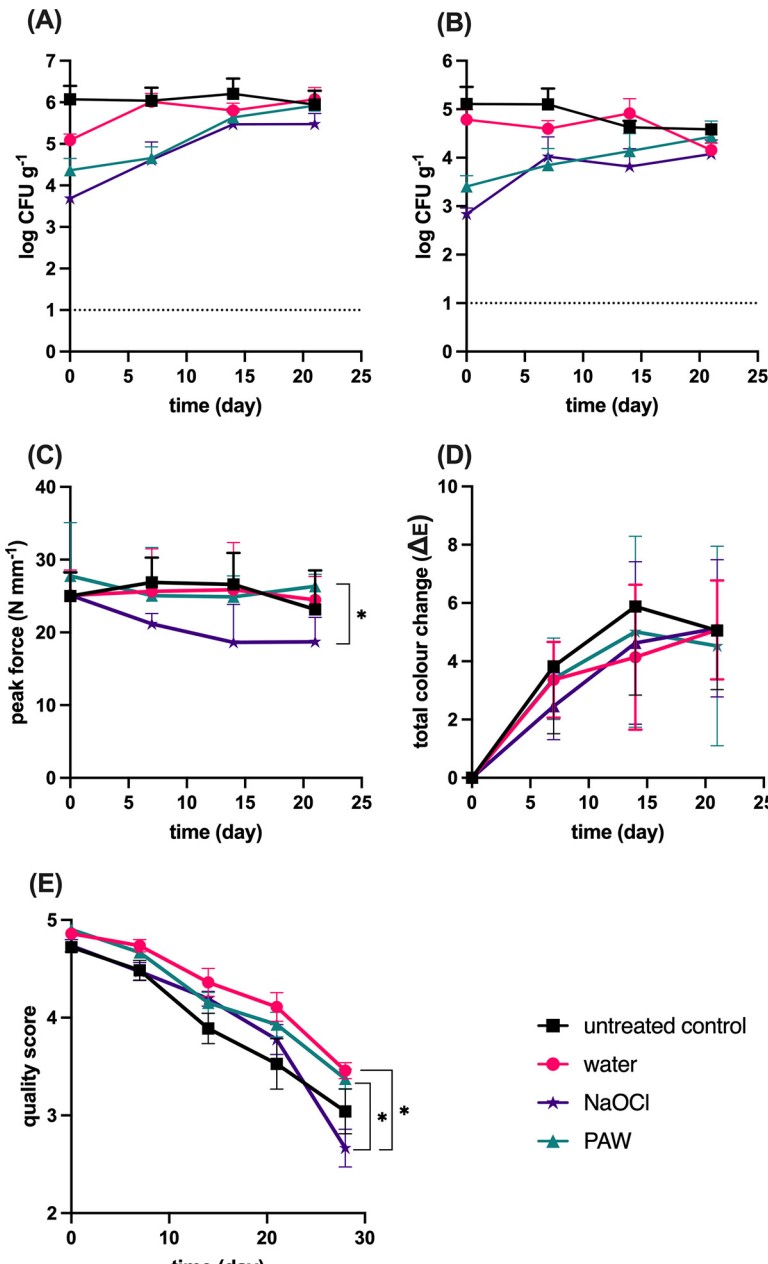

**FIG 5** Quality parameters of cucamelons during storage following different wash treatments. (A) Counts of total viable mesophilic bacteria; (B) counts of total yeast and molds; (C) texture of the cucamelons, with higher peak force units representing a firmer fruit; (D) total color change of the surface of the cucamelons over time with the light and dark sections combined; (E) quality of cucamelons as scored by a panel marking organoleptic properties. P values of <0.05 are denoted by *, and error bars represent standard error of the mean.

lower and were on average below 3, indicating that the product was no longer within acceptable specifications for consumption (Fig. 5E).

**Plasma chemistry modeling to determine $\cdot O_2^-$ production.** The electrical voltage and current characteristics of the DBDD plasma were determined in order to model the plasma chemistry and $\cdot O_2^-$ production of the system. The measured voltage, current response, and the calculated power from the DDBD reactor are shown in Fig. S1 in the supplemental material. The peak voltage and current were 8 ($\pm0.1$) kV and 0.12 ($\pm0.1$) A, respectively, and the estimated average power dissipated to the plasma discharge was approximately 12 ($\pm0.24$) W.

The plasma discharge was complex and varied over time (Fig. S1); therefore, the *E/N*

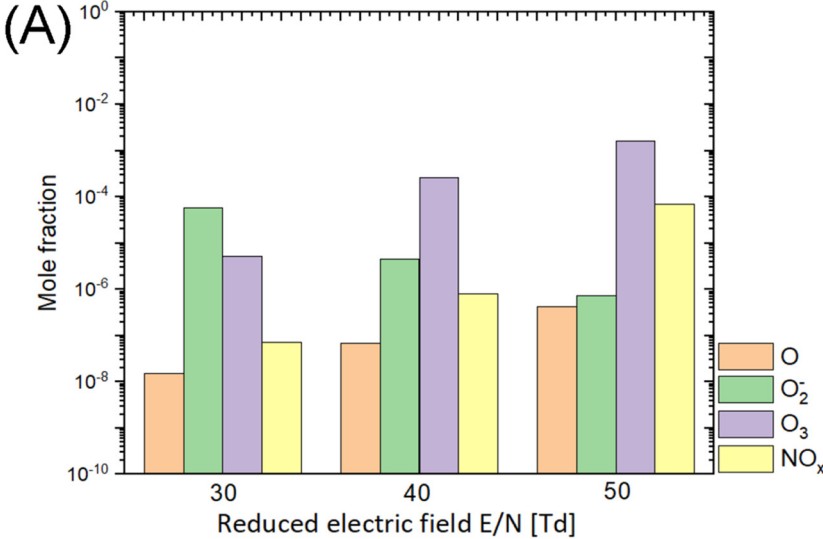

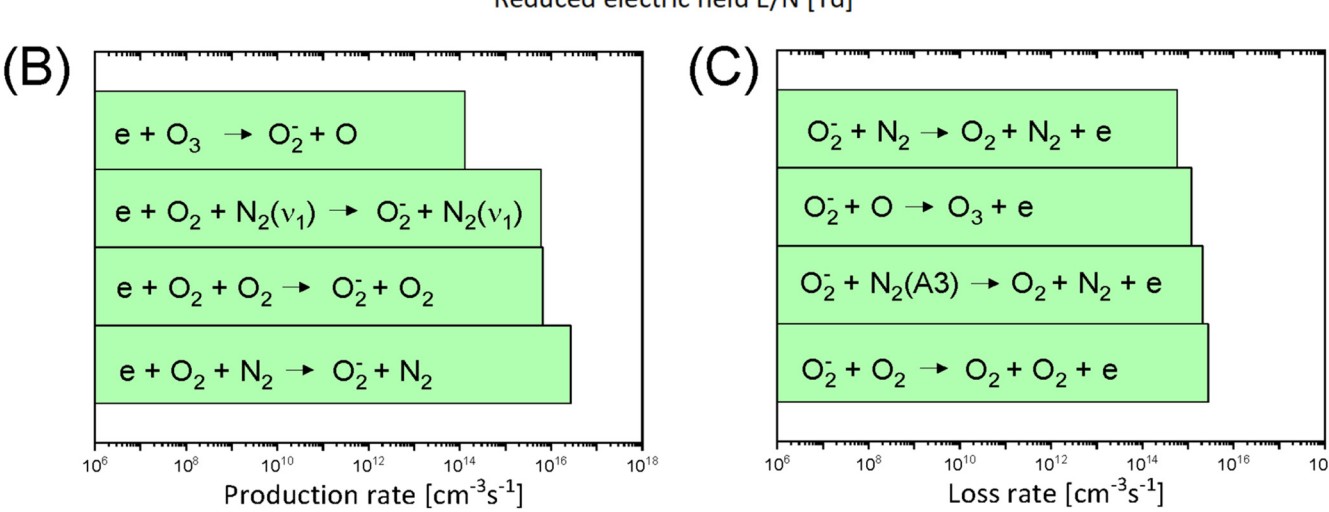

**FIG 6** Plasma chemistry modeling of the $N_2/O_2$ plasma discharge at atmospheric pressure. (A) The mole fraction of important gas species produced by the DBDD plasma under $E/N$ conditions of 30, 40, and 50 Td. $NO_x$ indicates total nitrogen oxide species in the gas phase including NO, $NO_2$, $NO_3$, $N_2O_3$, $N_2O_4$, and $N_2O_5$. (B and C) The important production (B) and loss (C) mechanisms and rates of $O_2^-$ in the gas phase. The settings used for this modeling were a glass transition temperature of 300 K, a residence time of 0.067 s, and a gas composition of 0.8:0.2 $N_2$:$O_2$.

values of 30, 40, and 50 Townsends (Td) were simulated for the initial plasma chemistry modeling. Figure 6A shows the kinetic modeling result with the predicted mole fraction of important gas species in the air discharge under these $E/N$ conditions. At 30 Td, $\cdot O_2^-$ was predominant compared to the other reactive species in the air discharge. At the higher $E/N$ values, the production of $O_3$ and $NO_x$ species increased whereas $\cdot O_2^-$ production decreased. This was not reflective of the DBDD-PAW physicochemical properties defined by our previous analysis (30) or the findings of the scavenger assays with Tiron shown in Fig. 3; therefore, 30 Td was the assumed $E/N$ value used for subsequent modeling.

The plasma modeling results in Fig. 6B and C depict the production and loss mechanism of $\cdot O_2^-$ in the DBDD system. Three-body attachment was the main pathway to generate $\cdot O_2^-$ in this model. This occurs when an electron attaches to the oxygen molecule to make $\cdot O_2^-$ and a third, neutral molecule such as $N_2$ or $O_2$ is present to absorb the released energy and complete the reaction.

$$e^- + O_2 + M\,(3rd\,body,\,any\,neutrals) \rightarrow \cdot O_2^- + M$$

In principle, this electron attachment process is most likely to occur at a low electron energy (<0.1 eV) due to the low threshold energy requirement as shown in

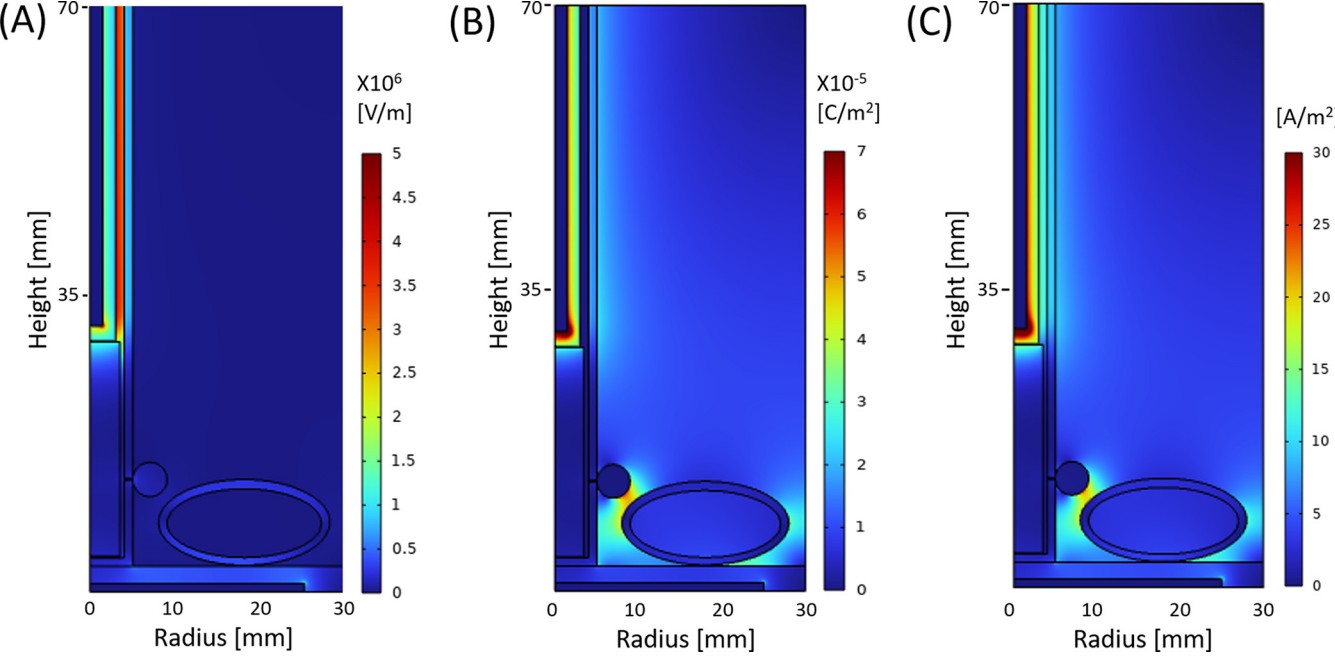

**FIG 7** Modeling of the spatial distribution of the normal component of electric field (A), polarization (B), and maximum current density (C) at the peak voltage, 8 kV, in the entire configuration of the DDBD reactor.

Fig. S2. However, a high electron density is required to increase the probability that this reaction will occur. The electron density is mainly determined by the $E/N$ in a given reactor configuration. Therefore, operating in the optimum range of $E/N$ and electron density is crucial for the effective production of $\cdot O_2^-$ with this DBDD system. Under higher $E/N$ conditions, there is a greater density of dissociated oxygen atoms and excited nitrogen molecules (Fig. 6A), resulting in a high loss rate of $\cdot O_2^-$ (Fig. 6C) and contributing to the competing production of $O_3$ or $NO_x$ (Fig. 6B). Taken together, these results show that lower $E/N$ is required for efficient $\cdot O_2^-$ production.

**Modeling of the electric field and its effect on bacterial cells.** Figure 7 depicts the spatial distribution of the electric field and the polarization field in the DBDD reactor model at the peak applied voltage of 8 kV and 60-kHz alternating current (AC) input. As shown in Fig. 7A, a strong electric field is formed in the discharge gap between the glass sheaths of the high-voltage electrode, whereas in the rest of the dielectric domain, including the water and the cucamelon, the electric field is significantly attenuated. Figure 7B and 7C show the polarization field and maximum current density that are generated, respectively, which are particularly enhanced near the interface between the bubble and the cucamelon surface. A more detailed view of the local electric field and the positioning of the bacterial cells in relation to a plasma bubble and the cucamelon is shown in Fig. 8A. Figures 8B and C show the electric field distribution of bacterial cells floating in solution or attached to the cucamelon surface, respectively. These demonstrate that a high local electric field of over 2.0 kV/cm can be formed inside the bacterial cell under these conditions. This is strongly dependent on the relative position of individual bacterial cells; when the cucamelon is located at the top of the water, the maximum local electric field experienced by the bacteria is much lower at 0.24 to 0.55 kV/cm (Fig. S3). A highly enhanced local polarization field is observed at the tips of bacterial cells in solution (Fig. 8D) and between bacterial cells and the cucamelon skin (Fig. 8E). The largest induced polarization current density value was over 40 A/m² in this model. This modeling is reflected in the pattern of damage to PAW-treated *L. monocytogenes* cells seen in the SEM images (white arrow, Fig. 4), indicating a possible influence of this strongly enhanced polarization current at the ends of the bacterial cells that leads to cell damage and ultimately cell death.

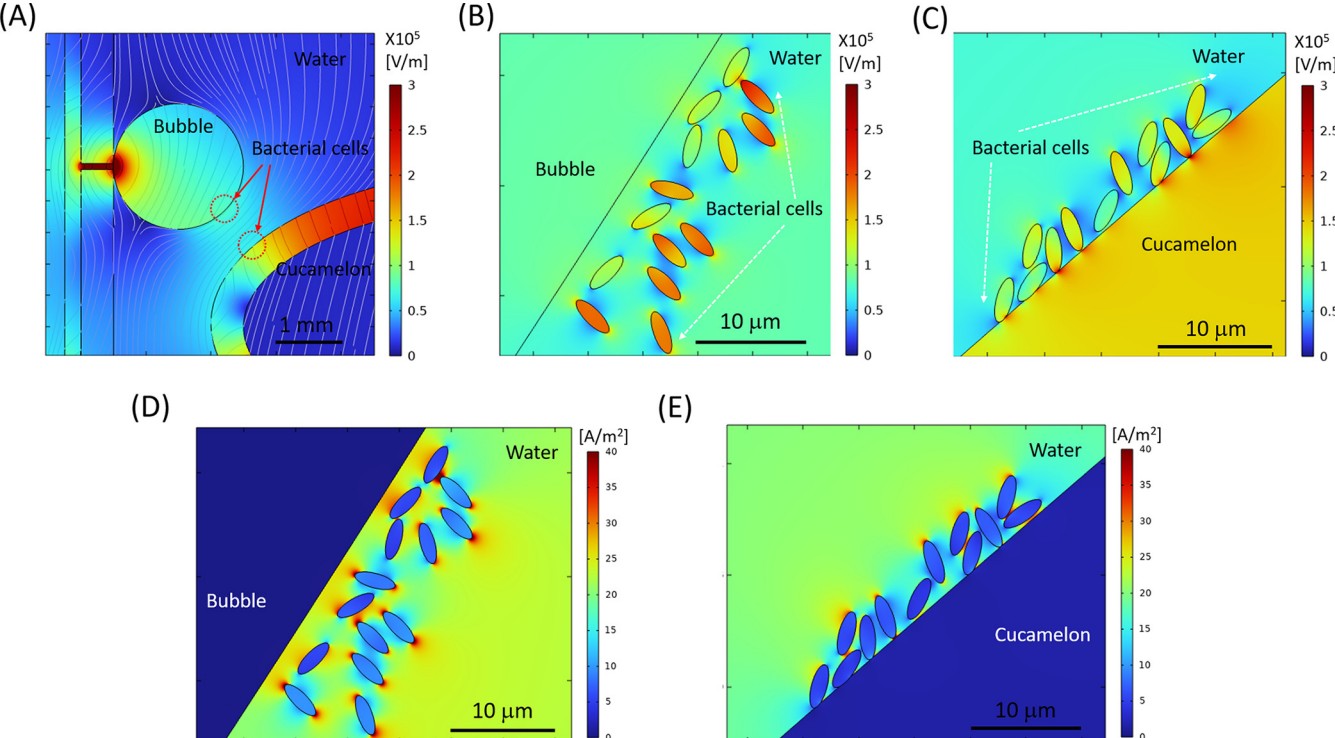

**FIG 8** Modeling of the electric field distribution and charge accumulation for bacterial cells in the DBDD-PAW system. (A) Electric field analysis, where gray lines show the streamline of the electric field between the bubble and the cucamelon at peak voltage 8 kV. (B and C) Detailed local electric field distribution close to the bacterial cells shown next to the bubble (B) and on the cucamelon skin (C). (D and E) Simulated maximum current density distribution near bacterial cells floating in the water close to the bubble (D) and attached on the cucamelon skin (E).

## DISCUSSION

This study evaluated the sanitization efficacy of an *in situ* PAW reactor in comparison to the established sanitizer NaOCl, using a cucurbit fresh produce model. We observed that treatment with PAW preserved the shelf life and quality of the cucamelons while effectively reducing microbial loads of the background microbiota as well as inoculated pathogenic bacteria. Although NaOCl was slightly more effective than PAW at reducing the counts of the pathogenic Gram-negative bacteria on the cucamelon surface, this sanitizer resulted in produce that was lower in quality over time. The mechanisms behind the acute antimicrobial power of the plasma system used in this study were explored, with $\cdot O_2^-$ shown to be an essential reactive species. Electric field and charge accumulation modeling demonstrated that bacteria on cucamelon surfaces and in the wash water experience strong electrical forces, which may act synergistically with the reactive species such as $\cdot O_2^-$ to produce powerful antibacterial activity.

There are growing calls to increase the consumption of fresh produce and to reduce food wastage; however, this can be at odds with food safety requirements where strong sanitizing agents may be needed to kill pathogens and reduce microbial load. In the current study, we used a cucamelon model in a small-scale DBDD plasma reactor and demonstrated that DBDD-PAW is a competitive sanitizer technology for fresh produce. A 2-min treatment with either DBDD-PAW or NaOCl reduced 6-log CFU mL$^{-1}$ bacterial pathogens in the wash water to below detectable limits. Furthermore, the *in situ* bubbling DBDD-PAW system used in the current study induced a substantially more rapid antimicrobial effect than did previous PAW models, including a DBD electrode positioned above the water that required 40 min of treatment (32) or a remote DBD bubble system that required pre-activation of the water for more than 80 min (33), making this system more feasible for industrial use.

Bacteria that were adhered to the surface of the cucamelons were also significantly reduced by DBDD-PAW treatment. Pathogenic species *E. coli*, *Salmonella enterica*, and

*L. monocytogenes* inoculated on the cucurbit surface were all reduced by 3 log CFU g$^{-1}$ with DBDD-PAW compared to the unwashed control. A similar result has been reported using lettuce leaves and a bubbling DBD-PAW system, where a 2-log CFU g$^{-1}$ reduction of *Listeria innocua* occurred after 3 min of treatment (27). NaOCl treatment was slightly more effective than PAW at reducing the Gram-negative pathogens on the cucamelon surface; however, the two treatments were similarly effective in reducing *L. monocytogenes* populations. Longer preactivation and treatment times may be required to optimize the antimicrobial power of PAW; however, this has challenges for steady-state operation and can reduce the quality of the fresh produce, as demonstrated in a recent study with kale and spinach (34). Overall, PAW treatment was highly effective at reducing the counts of pathogenic bacteria and demonstrated antimicrobial efficacy competitive with that of NaOCl treatment in a rapid 2-min wash time.

Resident mesophilic fungi and bacteria are associated with postharvest spoilage of fresh produce (35), and these were rapidly reduced by the PAW treatment, giving results similar to treatment with NaOCl (Fig. 5A and B). Comparable reductions have been reported for PAW treatment of baby spinach and rocket leaves and for bean sprouts (36–38), indicating application across a range of produce types. In addition, treatment with PAW maintained organoleptic quality and increased the shelf life of cucamelons over time in comparison to NaOCl. Enhanced textural quality following PAW has been demonstrated previously for button mushrooms (25), apples (39), and Chinese bayberries (40), and previous studies have reported that PAW treatment does not significantly alter the color of grapes (23), lettuce (41), and spinach (37). Together, these findings indicate the capacity for DBDD-PAW treatment to reduce both pathogens and mesophiles while extending shelf life and maintaining a high quality of the produce, indicating better performance overall in comparison to the established sanitizer, NaOCl.

Modeling suggested that PAW treatment caused a highly localized surface charge density on the poles of the bacterial cells, which was supported by the SEM analysis showing the PAW-treated bacteria ruptured at their ends. This dramatic physical disruption is very similar to the appearance of cells following pulsed electric field treatment (42) and was quite distinct from NaOCl treatment, where crumpling and puckering of the cell surface were observed. At and near the interface of different materials such as water/bacterial cell or bacterial cell/cucamelon skin, a high-density surface charge can be generated by the dipole-like response of dielectric material, and this leads to a strong local polarization and electric field. The induced current density at the interface of bacterial cells and water was found to reach a very high value of over 4.0 mA/cm$^2$. Membrane damage and leakage of bacterial cells have been observed in a much lower current range of 50 to 80 $\mu$A/cm$^2$, although this was after a longer treatment time of 30 min and under direct current conditions (43). Regardless, charges of opposite polarity in membranes oscillate according to the applied electric field, leading to a strong polarization field and induced currents. We hypothesize that this physical effect may have caused the observed membrane stresses and pore formation, especially where the highly localized surface charge density was accumulating at the ends of the rod-shaped cells.

The superoxide anion ($\cdot O_2^-$) has previously been demonstrated to be critical for the antimicrobial activity of the DBDD-PAW (30), and this was confirmed in the current study where the addition of Tiron, an $\cdot O_2^-$ scavenger, significantly reduced the antimicrobial activity of the DBDD-PAW. $\cdot O_2^-$ has a short half-life and a negative charge when in a solution with a neutral pH (44), which would typically prevent it from passing through the membrane of bacterial cells (45). Therefore, while $\cdot O_2^-$ is required for antimicrobial power, this reactive species on its own may not be sufficient for the activity observed. Previous studies have demonstrated that the $\cdot O_2^-$ produced in a DBDD-PAW system is a precursor to highly antimicrobial reactive species such as hydroxyl radicals and singlet molecular oxygen, and these secondary reactive species have been shown to contribute to the potent antimicrobial effects of PAW on yeast (46) and *Salmonella enterica* serovar Typhimurium (47). Therefore, $\cdot O_2^-$ and/or downstream reactive

species, combined with the physical effects of the *in situ* plasma treatment that result in membrane damage and permeabilization, are likely synergizing to create the powerful antimicrobial effects produced by DBDD-PAW.

PAW technologies that rely on $\cdot O_2^-$ may be more cost-effective and more readily applied to industrial applications than those relying on other reactive oxygen and nitrogen species (RONS) for the *in situ* treatment of fresh produce. As an ionic species, $\cdot O_2^-$ can dissolve directly into water without loss, unlike other RONS species such as NO or $O_3$ that have a low solubility in water (48). This rapid solvation of $\cdot O_2^-$ means preactivation or long treatment times are not required for DBDD-PAW, unlike PAW systems where less-soluble RONS provide the antimicrobial power (30). In addition, the formation and accumulation of $\cdot O_2^-$ occur at a lower electron energy and $E/N$ than those of other RONS (Fig. 6; see also Fig. S2 in the supplemental material) (49), and the power requirements of the DBDD are very low at only 12 W (Fig. S1). Finally, the DBDD-PAW system does not increase the temperature of the water substantially or produce large concentrations of longer-lived reactive species such as nitrates and nitrites that have been seen with a spark discharge PAW reactor (30), so there are no requirements for cooling systems and fewer concerns with chemical residues on the fresh produce.

The findings presented in this study indicate that the DBDD-PAW system is a competitive sanitizer technology that warrants upscaling for postharvest treatment of fresh produce. This system may be advantageous over traditional chemical sanitizers in terms of sustainability and cost as high antimicrobial power is achieved with only air, tap water, the plasma reactor, and small amounts of electricity (12 W), and there is no need to dispose of hazardous effluents. PAW treatment extended the quality of the fresh produce to a greater extent than NaOCl, and the DBDD-PAW system used here reduced pathogenic bacteria more rapidly and without the preactivation step often required by other PAW systems (32, 33). However, the activity of DBDD-PAW relies on the short-lived reactive species $\cdot O_2^-$ and the effects of an electric field and will be most effective when applied in an *in situ* wash system. There are important safety considerations that also need addressing, including the potential for electrical hazards and the possible buildup of ozone during plasma production. For future scale-up of this technology, reactor design and the position of the electrodes in the wash systems must be considered to guarantee effective antimicrobial power and thereby maximize food safety.

## MATERIALS AND METHODS

**Sanitizer and wash preparations.** Three wash treatments were used in this study: a sterile tap water control, 100-ppm total chlorine NaOCl solution, and PAW. An untreated control, where cucurbits did not receive any washing, was also included. All treatments were made using autoclaved tap water cooled to 4°C in a final volume of 200 mL. The chemical analysis of the Sydney tap water used in this study has been published previously (30). Concentrated NaOCl (Sigma-Aldrich) was diluted to 100 ppm (±1 ppm) total chlorine using a Kemio test kit with test sensors suitable for high-range chlorine concentrations (Palintest, Tyne & Wear, United Kingdom). The pH was adjusted to 6.5 ± 0.1 (SevenCompact S220; Mettler-Toledo) using 10% lactic acid. The PAW system configuration is illustrated in Fig. 1. PAW was generated with a DBDD probe (PlasmaLeap Technologies), and power was supplied from a Leap100 micropulse generator (PlasmaLeap Technologies). The power supply settings used were based on the findings of our previous study (30). These settings included a frequency of 1,500 Hz, a voltage of 120 V, a duty cycle of 100 microseconds, and a resonance frequency of 60 kHz. Compressed air at a flow rate of 1 standard L per min (SLM) was used as the processing gas.

**Bacterial culture preparation.** The cultures listed in Table 1 were stored as glycerol stocks at −80°C. Prior to experimentation, bacteria were resuscitated from frozen stocks by plating onto either tryptic soy agar (TSA; 17 g L$^{-1}$ pancreatic digest of casein, 5 g L$^{-1}$ papaic digest of soybean meal, 5 g L$^{-1}$ sodium chloride, 15 g L$^{-1}$ agar-agar) for *S. enterica* and *E. coli* with incubation at 37°C for 24 h or tryptic soy sheep blood agar (TSBA; TSA with 5% defibrinated sheep's blood) for *L. monocytogenes* with incubation at 30°C for 48 h. A single colony of each strain was then inoculated into separate tubes with 10 mL of tryptic soy broth [TSB; 17 g L$^{-1}$ pancreatic digest of casein, 2.5 g L$^{-1}$ D-(+)-glucose monohydrate, 3 g L$^{-1}$ papaic digest of soybean meal, 5 g L$^{-1}$ sodium chloride, 2.5 g L$^{-1}$ dipotassium hydrogen phosphate] for 18 h with shaking at 200 rpm at 37°C for *S. enterica* and *E. coli* or 30°C for *L. monocytogenes*. Cultures were centrifuged at 3,000 rpm for 10 min at 4°C, resuspended in phosphate-buffered saline (PBS; Oxoid), stored at 4°C, and used within 4 h. Immediately prior to experimentation, equal volumes of the three strains of each species were mixed to create a 3-strain cocktail. Each cocktail was serially diluted in PBS, spread plated, and incubated as described above to determine the final inoculum concentrations.

**TABLE 1** Bacterial isolates used in this study

| Species | Strain designation[a] | Serotype | Source (yr), details |
|---|---|---|---|
| *Salmonella enterica* subsp. *enterica* | ICPMR: 06-17-184-1802 | Saintpaul | Feces (2017) |
| *S. enterica* subsp. *enterica* | ICPMR: 80-17-173-5603 | Hvittingfoss | Feces (2017), clustered with a 2016 rockmelon salmonellosis outbreak in Australia |
| *S. enterica* subsp. *enterica* | ICPMR: 80-17-149-5555 | Anatum | Feces (2017), clustered with a 2016 bagged salad product salmonellosis outbreak in Australia |
| *Escherichia coli* | ICMPR: 40-16-302-2227 | O157:H7 | Feces (2016) |
| *E. coli* | ICPMR: 80-16-270-5374 | O26:H11 | Feces (2016) |
| *E. coli* | ICPMR: 80-16-302-4575 | O111:H⁻ | Feces (2016) |
| *Listeria monocytogenes* | ATCC: 51772; 3M: 4395 | 1/2a | Cheese |
| *L. monocytogenes* | ICMPR: 80-13-220-4103 | 1/2b | Blood (2013), same binary type as 2010 fresh-cut melon listeriosis outbreak in Australia |
| *L. monocytogenes* | ICPMR: 80-18-038-5080 | 4bV | Blood (2018), clustered with a 2018 rockmelon listeriosis outbreak in Australia |

[a]ICPMR, Institute of Clinical Pathology and Medical Research, University of Sydney.

**Inoculation, treatment, and microbial analysis of cucamelons.** Fresh unwashed cucamelons were homegrown and harvested a day before experimentation and were stored at 9°C with 85% humidity. For each treatment condition, two cucamelons of similar size each weighing a total of 10 g ($\pm$0.5 g) were selected. Cucamelons were first briefly immersed in 80% ethanol and rinsed with sterile tap water to reduce background microbial load. Each cucamelon was then spot inoculated with $10 \times 10$ $\mu$L (100 $\mu$L total) of the *E. coli* or *S. enterica* inoculum at a final concentration of $1 \times 10^9$ CFU mL$^{-1}$. Cucamelons were spot inoculated with $20 \times 10$-$\mu$L spots (200 $\mu$L total) of *L. monocytogenes* at $8 \times 10^8$ CFU mL$^{-1}$, as this species has a lower adhesion to cucumber (50). Inoculated cucamelons were dried in a biosafety cabinet until there was no visible moisture remaining (approximately 45 min). To simulate contaminated wash water, 200 mL of the inoculum was added to each of the 200-mL treatment solutions along with the inoculated cucamelons. For the PAW condition, the cucamelons and contaminated wash solution were plasma treated for 2 min as described under "Sanitizer and wash preparations." To standardize the effect of the bubbling across the treatments, the DBDD probe was inserted into the water and bubbled without plasma generation for the control and NaOCl conditions. The probe was sanitized between treatments by wiping with ethanol and then rinsing with sterile tap water.

Following treatment, the cucamelons were removed with sterile tweezers, placed into a stomacher filter bag with 40 mL of PBS, and homogenized with a paddle blender (BagMixer 400; Interscience, France) for 2 min. An untreated control was included in which inoculated cucamelons were homogenized without any wash treatment. One milliliter of cucamelon homogenate and 1 mL of the wash water were serially diluted with PBS and spread plated onto TSA for *E. coli* and *S. enterica* or BA for *L. monocytogenes* and incubated as described under "Bacterial culture preparation" for enumeration of pathogens.

Our previous work had identified $\cdot O_2^-$ as a critical reactive species for the antimicrobial power of the DBDD-PAW system (30). We investigated the antimicrobial role of $\cdot O_2^-$ in the fresh produce model by adding the $\cdot O_2^-$ scavenger Tiron (Sigma-Aldrich) to the PAW system at a final concentration of 20 mM (51).

All experiments were performed in duplicate, with three biological replicates performed on separate days. Cucamelons were harvested every fortnight across the season for each biological replicate. To identify significant differences between treatment groups, one-way analysis of variance (ANOVA) with Tukey's multiple-comparison tests was performed using GraphPad Prism version 8.0.0 (GraphPad Software). A $P$ value of <0.05 was considered statistically significant.

**SEM.** Scanning electron microscopy (SEM) was performed to evaluate morphological changes to bacterial cell structures after treatment with PAW, NaOCl, or the water control. Cucamelons were inoculated with *E. coli* or *L. monocytogenes* and treated as described above. Immediately after treatment, a sterile scalpel was used to slice 1-mm-thick sections from the cucamelon surface. The sections were placed into 2.5% glutaraldehyde fixative solution in 0.1 M phosphate buffer (pH 7) at room temperature for 1 h with gentle agitation and then stored at 4°C. Samples were then washed with 0.1 M phosphate buffer 3 times for 5 min. The samples were then dehydrated using an ethanol concentration gradient with 2 washes for 5 min in 30, 50, 70, 80, and 90% ethanol followed by three 5-min washes in 95 and 100% ethanol. Next, the samples were dried using critical point drying with liquid $CO_2$. Samples were then fixed to aluminum stubs with carbon tape and sputter coated with 10 nm of gold at 39 mA using a CCU-010 HV high-vacuum compact coating system (Safematic, Switzerland). Samples were imaged using the Zeiss Sigma VP HD scanning electron microscope at 5 kV (Zeiss, Germany).

**Shelf life and organoleptic quality assessment of cucamelons following treatment.** The effects of each wash treatment on the background microbiota, organoleptic quality, and shelf life of the cucamelons were assessed. For each of the four conditions, six cucamelons with no visible defects were selected. The cucamelons were treated with the sanitizers as described above with the inoculation and ethanol rinse steps omitted. The cucamelons were then placed into 6-well plates (Corning Costar) and stored at 9°C with 85% humidity (52). The following experiments were repeated across three biological replicates performed on different weeks.

**Assessment of the microbial load on cucamelons as a function of treatment.** On days 0, 7, 14, and 21, two cucamelons from each treatment group were removed and stomached as described above. One milliliter of the cucamelon homogenate was serially diluted in PBS, spread on plate count agar (PCA; 5 g L$^{-1}$ enzymatic digest of casein, 2.5 g L$^{-1}$ yeast extract, 1 g L$^{-1}$ glucose, 15 g L$^{-1}$ agar-agar), and incubated at 25°C for 3 days to enumerate the total mesophilic aerobic bacteria. One milliliter of the cucamelon homogenate was also serially diluted in PBS, spread onto dichloran rose bengal chloramphenicol agar (DRBC; Oxoid), and incubated at 25°C for 5 days to enumerate yeasts and molds.

**Texture analysis.** The flesh firmness of the treated cucamelons was analyzed using a TMS-Pro texture analyzer (Food Technology Corporation, VA, USA) fitted with a 3.0-mm-diameter cylindrical probe. The probe was programmed to descend at a speed of 500 mm min$^{-1}$ to a distance of 5 mm. Intact cucamelons were positioned under the probe so that they were punctured approximately at the fruit equator. Flesh firmness was estimated as the peak force (newtons) measured during compression. Duplicate cucamelons from each wash treatment were tested on day of treatment and after 7, 14, and 21 days of storage.

**Color measurement.** Change to the color of cucamelon skin on day of treatment and following 7, 14, and 21 days of storage was quantified using image analysis. Cucamelons were imaged with a stereomicroscope (SZM-45B2; Optex) and microscope camera (5 MP Microscope USB camera; Westlab). The stage was illuminated using an LED lighting panel (AL-F7; Aputure). Three photos covering random areas of each cucamelon were taken by gently rotating the fruit using sterile tweezers. Six cucamelons were photographed for each wash treatment, and the experiment was repeated over three biological replicates performed on different days. To account for any changes in ambient lighting conditions, the cucamelons were photographed on the same white background. The color values of the background of the photos in the linear RGB color space were then standardized across all photos using R (53).

As the cucamelon surface is patterned with sections of light and dark green, these were analyzed separately using computer vision and statistical clustering methods. The sample images were first cropped to contain only the area of the melon itself. The "superpixel" algorithm (54) was implemented as part of the OpenImageR package (55) and is a computer vision algorithm that determines groups of contiguous pixels based on their proximity and divides them into 300 sections. To classify each group identified by the superpixel algorithm as light or dark, the k-means clustering algorithm was applied on the median L*a*b* color channel values. In this CIELab color space system, the L-axis, a-plane, and b-plane detail the level of brightness, green/red, and blue/yellow of a section, respectively.

For each sample, the average for each of the L*a*b* color channels was identified individually for the light and dark sections. Total color difference, Delta E ($\Delta E$*), was calculated by the following equation where $_1$ and $_2$ indicate the values on day 0 and day 21, respectively:

$$\Delta E_{ab}^* = \sqrt{(L_2^* - L_1^*)^2 + (a_2^* - a_1^*)^2 + (b_2^* - b_1^*)^2}$$

The browning index (BI) indicates the brown color intensity of an image (56) calculated as follows:

$$\text{BI} = 100\frac{x - 0.31}{0.172}; \ x = \frac{(a^* + 1.75 \ L*)}{(5.645 \ L^* + a^* - 3.012 \ b^*)}$$

The statistical significance of color change between day 0 and day 21 was tested for both light and dark sections independently and for the two sections combined. Pairwise $t$ tests between each treatment group were performed using R. One-way ANOVAs were also completed on the treatment type to see if treatment was a significant predictor of change between day 0 and day 21.

**Sensory quality assessment.** The effect of sanitizer treatments on the quality of cucamelons over 4 weeks of storage was evaluated using a sensory evaluation by 6 panelists. The acceptability of the product was based on the freshness, appearance, deterioration, and uniformity. This qualitative assessment was scored with a 5-point rating scale described as follows: 1, extremely poor quality, inedible and with unacceptable appearance; 2, poor quality, excessive defects and not usable; 3, low quality, moderately objectionable defects; 4, good quality with some defects; 5, excellent. Scores of 3 and lower indicated that the product was no longer within acceptable specifications for consumption. Six cucamelons were assessed from each treatment, and the experiment was performed twice on different weeks.

**Supporting plasma chemistry model and electric field analysis.** To model the plasma chemistry and electric fields, the gas residence time, voltage, and current parameters of the DBDD system were first investigated. The discharge volume was estimated to be 1.1 cm$^3$ based on the reactor's dimensions, which included an outer radius of 0.4 cm, an inner radius of 0.3 cm, and a height of 5 cm. By dividing the discharge volume by the gas flow rate, the residence time of gas species within the discharge volume was determined to be 0.067 s at the gas flow rate of 1 SLM. The voltage and current characteristics of the DDBD reactor were measured using a digital oscilloscope (DS61040; Rigol) with a high-voltage probe (PVM-6; North Star) and a current sensor (HET10AB15U10; PEMCH Tech.). The experimental configuration was as shown in Fig. 1, including voltage and current measurement setup. The reduced electric field ($E/N$) is calculated by dividing the electric field strength ($E$) by the density of neutral gas molecules ($N$) in the plasma. The $E/N$ ratio is related to the energy distribution of electrons in the plasma, and it is an important parameter for plasma modeling. However, plasma exhibits highly transient and nonuniform characteristics, as depicted in Fig. S1 in the supplemental material. Therefore, an averaged $E/N$ value that accounts for these spatial and temporal variations was estimated using the QV Lissajous plot technique (57). The estimated $E/N$ was approximately 30 Td as shown in Table S1, and so $E/N$ values of

**TABLE 2** Gas phase considered in the $N_2/O_2$ plasma chemistry model

| Category | Chemical species |
|---|---|
| Ground-state molecules and radicals | $N_2$, $O_2$, $O_3$, NO, $NO_2$, $NO_3$, NO, $NO_2$, $NO_3$, $N_2O$, $N_2O_3$, $N_2O_4$, $N_2O_5$ |
| Vibrationally excited molecules | $N_2(v_i, i = 1–8)$, $O_2(v_i, i = 1–4)$ |
| Electronically excited molecules | $N_2(A3)$, $N_2(B3)$, $N_2(a'1)$, $N_2(C3)$, $O_2(a1)$, $O_2(b1)$ |
| Atoms | N, N(2D), N(2P), O, O(1D), O(1S) |
| Ions | $N^+$, $N_2^+$, $N_3^+$, $N_4^+$, $O^+$, $O_2^+$, $NO^+$, $N_2O^+$, $O^-$, $O_2^-$ |

30, 40, and 50 Townsends (Td) were used for the initial plasma chemistry simulation. The influence of these parameters on the production of $\cdot O_2^-$ and other important gas products listed in Table 2 was investigated. As the gas composition at 30 Td was closest to that determined empirically in our previous study (30), we used 30 Td as the assumed $E/N$ value used for subsequent plasma simulations.

A model of the plasma chemistry was used to investigate the pathways of reactive species production in the DBDD system. Based on a previous $N_2$-$H_2O$ plasma discharge model (58), a $N_2/O_2$ plasma chemistry model was developed that included more oxygen-related reactions to enable important gas products and predominant reaction pathways of $\cdot O_2^-$ to be investigated (Table S2). Open-source ZDPlasKin (59) combined with the Boltzmann equation solver BOLSIG+ (60, 61) was used to provide the reaction coefficients for different electron interactions, including electron attachment, ionization, vibrational and electronic excitation, dissociation, and important chemical reactions between $N_2$ and $O_2$, as listed in Tables S2 and S3. The gas-phase chemical reactions between nitrogen and oxygen species were mostly adapted from a previous study (62). The gas-phase reactions included in the kinetic input data were as presented in the reaction equation 1. These were converted into a coupled differential equation form of particle conservation, equation 2, for individual species $i$, which included different production and loss reactions:

$$aA + bB \rightarrow a'A + cC \tag{1}$$

$$\frac{d[N_i]}{dt} = \sum_{j=1}^{j_{max}} K_{ij}(t) \tag{2}$$

where $[N_i]$ is the density of species $i$, $K_A = (a' - a)k$, $K_B = -bk$, $Kc = ck$, $K_j = k_j[A]^a[B]^b$, and $k_j$ indicates the reaction coefficient of reaction 1.

The Electrostatic Interface of COMSOL Multiphysics (V6.0) was used to simulate the spatial and temporal electric potential and field distribution with the peak voltage of 8 kV and 60-kHz AC applied at the high-voltage electrode as shown in Fig. S1. A two-dimensional (2D) axisymmetric model of the DBDD was taken to reduce the calculation time. The geometry of the reactor is shown in Fig. 2A. The dimensions were defined as a 2-mm-diameter high-voltage (HV) electrode sheathed by a 1-mm-thick glass tube, a 1-mm discharge gap, and a 1-mm-thick outer glass tube. The gas bubbles were defined by a 3-mm diameter filled with air. The cucamelon was simulated at two positions: at the bottom of the water as shown in Fig. 2A and at the top of the water as shown in Fig. S3. The cucamelon was modeled as having 7 or 14 bacterial cells on its surface or floating in bulk water as shown in Fig. 2B and C and Fig. S3. The bacteria were modeled with a long-axis radius of 2 $\mu$m and a short-axis radius of 0.7 $\mu$m, presented in a simplified ellipse shape. The relative permittivity ($\varepsilon/\varepsilon_0$) of cucamelon skin was set as 3, as per a previous study (63) that assigned this value for apple skin. For the bacterial cell, a higher value of 15 was taken (64). The minimum and maximum size of the COMSOL mesh geometry was set as 0.1 $\mu$m and 0.5 mm, respectively, to provide enough detail within the bacterial cell region as shown in Fig. 2C. The normal component of the electric field ($E$) is calculated by ($\sqrt{E_r^2 + E_z^2}$), the normal polarization component ($P$) is calculated by ($\sqrt{P_r^2 + P_z^2}$), and the maximum current density ($J$) is calculated by ($\sqrt{J_r^2 + J_z^2}$), where the subscripts $r$ and $z$ indicate radial and axial components, respectively.

## SUPPLEMENTAL MATERIAL

Supplemental material is available online only.

**SUPPLEMENTAL FILE 1**, DOCX file, 1.6 MB.

## ACKNOWLEDGMENTS

We acknowledge the technical and scientific assistance of Sydney Microscopy & Microanalysis, the University of Sydney node of Microscopy Australia. We also acknowledge Nicholas Coleman for generously growing the cucamelons for this study.

This research was conducted within the Australian Research Council Industrial Transformation Training Centre for Food Safety in the Fresh Produce Industry (grant number IC160100025) funded by the Australian Research Council and industry partners from Australia and New Zealand and administered by the University of Sydney.

P. J. Cullen is the Chief Technology Officer of PlasmaLeap Technologies, the supplier of the plasma technology employed to generate plasma bubbles in this study.

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
