## [Reviewer comments · Microbiology Spectrum]

Microbiology Spectrum

An effective sanitizer for fresh produce production: In situ plasma activated water treatment inactivates pathogenic bacteria and maintains the quality of cucurbit fruit

Joanna Rothwell, Jungmi Hong, Stuart Morrison, Heema Vyas, Binbin Xia, Anne Mai-Prochnow, Robyn McConchie, Kim-Yen Phan-Thien, Patrick Cullen, and Dee Carter

Corresponding Author(s): Dee Carter, The University of Sydney

Review Timeline:

Submission Date:	January 3, 2023
Editorial Decision:	April 21, 2023
Revision Received:	June 13, 2023
Accepted:	June 18, 2023

Editor: Jasna Kovac

Reviewer(s): Disclosure of reviewer identity is with reference to reviewer comments included in decision letter(s). The following individuals involved in review of your submission have agreed to reveal their identity: Ki Ho Baek (Reviewer #4)

Transaction Report:

DOI: <https://doi.org/10.1128/spectrum.00034-23>

April 21, 2023

Prof. Dee A Carter
The University of Sydney
School of Life And Environmental Sciences
LEES Building F22, School of Life And Environmental Sciences
University of Sydney
Sydney, New South Wales 2006
Australia

Re: Spectrum00034-23 (An effective sanitizer for fresh produce production: In situ plasma activated water treatment inactivates pathogenic bacteria and maintains the quality of cucurbit fruit)

Dear Prof. Dee A Carter:

Link Not Available

Sincerely,

Jasna Kovac

Journals Department
Reviewer comments:

Reviewer #4 (Comments for the Author):

The submitted manuscript reports on the use of dielectric barrier discharge diffusor for decontamination of bacteria in solution and on the cucamelon surface. The author tried to design experiments considering the economic feasibility of the technology for industrial application. As an innovative approach in a demanding technological field, it results that the paper may deserves publication in the Microbiology Spectrum but after consideration of the 6 following comments.

C1 Introduction (line 64-68)

This is the part where the need for disinfection of fresh produce should be most emphasized to contribute to food safety. It is recommended to present the current status of food poisoning caused by fresh produce using the latest statistics.

C2 Materials and methods (Figure 1, line 145-147)

Unlike conventional PAWs, this device has a structure in which plasma is injected into water in real time. Therefore, the section where electrical discharge takes place is located very closely with water. Could this structure be a dangerous environment for workers?

C3 Materials and methods (line 181-184)

It is recommended to make the expression clearer.

[ten × 10 mL]: Is 100 mL of bacterial suspension (200 mL for *L. monocytogenes*) used in total?

C4 Materials and methods (Figure 1, line 188-189)

It is recommended to specify the position of the sample in the solution. In particular, the author explains that depending on the position of the sample in the solution, the electric field may be affected differently. In addition, the behavior of the sample may vary as bubbles enter the solution. As a result, the bactericidal effect of this device may be reduced in environments where the sample is located at the top of the solution. This means that the effect may vary depending on how many layers the sample is stacked in the solution. In addition, the effect may vary if the sample is light and floats in water.

C5 Results / Discussion (Figure 6, line 622-624)

The author explained that the amount of ozone dissolved in PAW is small due to the low solubility of ozone. Ozone is the most significant gas generated by a typical DBD system, and it is thought that a significant amount of ozone will also be generated in this system. Therefore, much of the ozone that is eventually insoluble in water can be exposed above the liquid. Please give us your opinion on this part and suggest alternative measures to the possibility of ozone exposure as the author considers industrial applicability important.

C6 Discussion (Figure 8, line 596-603)

Is there any way to directly observe how the electric field actually affected microbial cells? If the electric field is forming to a degree that has a profound effect on cells as the author suggests, I wonder if some cells should be killed even in an environment with tiron (-O₂- scavenger) added. Perhaps this is most similar to the environment in which only electrical discharge, other than reactive species, affects. How about observing cell membrane integrity after electrical discharge with the addition of tiron?

Reviewer #5 (Comments for the Author):

The manuscript by Rothwell et al. describes multidisciplinary research in the field of food science evaluating the use of plasma activated water for sanitizing cucamelons. It is the perception of this reviewer that the modelling components described in the current version would typically not be assessed or used by microbiologists. The modelling could have better served as an initial approach to optimize treatments so that a greater log reduction of pathogens was achievable. As presented the modeling is used to demonstrate mechanism. The ability of an audience of microbiologists to benefit from the modeling data is questionable. The description of the modeling activities is vague in the materials and methods section for a microbiology audience. It is unclear how the discharge volume was estimated (line 309), how the residence time was calculated (line 310), and how were the settings applied for the activated plasma were chosen. Also, how is the E/N ratio used? A number of gas products are considered in Table S1 but the text implies that only one of those is relevant in the system used. Please resolve this incongruence, is table S1 needed? Are nitrogen oxides produced in the plasma system chosen, possibly from N₂ in air? It seems that the rapid solvation of the oxygen species (line 629) works against the effectiveness of the treatment as the log reduction achieved is minimal and the produce spoils in about 10 days. Can the PAW be optimized to increase the shelf-life of the produce?

The modeling section should occur first in the materials and methods, so that it aligns with Figure 2 as an introductory piece. Although the protocols used for microbial counts, and texture, color and appearance evaluation may be standard, references are needed to support them in the materials and methods section.

A sentence should be added in line 132 about the use of tiron as a control treatment. It seems to come out of nowhere when the reader first encounters it in line 201. Tiron inclusion is working as a control treatment and should be presented as such.

Cucamelons, like other fruits and vegetables, are susceptible to equilibration with water in its surroundings (reference 32). Such dynamic needs to be considered regarding the use of solutions that were refrigerated (4C) prior to treatment application (line 170). The equilibration of fruits and vegetables with water depends on temperature. Was the temperature of the fruits during treatment measured? Were the solutions tempered?

Also, was there any indication in the electron microscopy work that the pathogens had penetrated the cucamelons' flesh? The effectiveness of the sanitizer to that of the plasma treatment may not be comparable as one treatment can reduce the microbes

in the 1mm outer flesh near the skin, but plasma cannot. Please elaborate on this aspect. (Mattos FR, Fasina OO, Reina LD, Fleming HP, Breidt F Jr, Damasceno GS, Passos FV. 2005. Heat transfer and microbial kinetics modeling to determine the location of microorganisms within cucumber fruit. J Food Sci 70(5):E324-E330.)

The use of tap water (line 103 & 139) and lactic acid (line 145): Was the mineral content of the tap water used evaluated? Would researchers be able to reproduce your findings with any tap water? Lactic acid (10% or 1.1 M) is an antimicrobial that was added to the chlorine solution for pH control. Thus, the sanitizer treatment could have been more effective than it typically is. Is lactic acid routinely used by processors in the wash waters for pH control?

Is there a reason for not studying the flavor of the treated and control samples?

Minor comments:

Line 190: How was the probe sanitized/sterilized between treatments?

Line 206: What was the definition of biological controls?

Line 231, 239 and throughout the manuscript: Please use the term microbiota instead of microflora (small plants).

Line 239: Assessment of the microbial load on cucamelons as a function of treatment.

Line 79: ...from food sanitation are hazardous...

Line 296 to 302: panelist

Line 423 to 425: Please show data for microbial counts from wash waters.

Line 662: Please spell out CTO

Staff Comments:

Preparing Revision Guidelines

Please return the manuscript within 60 days; if you cannot complete the modification within this time period, please contact me. If you do not wish to modify the manuscript and prefer to submit it to another journal, please notify me of your decision immediately so that the manuscript may be formally withdrawn from consideration by Microbiology Spectrum.

Report on Spectrum00034-23 by J.G. Rothwell et al

The submitted manuscript reports on the use of dielectric barrier discharge diffusor for decontamination of bacteria in solution and on the cucamelon surface. The author tried to design experiments considering the economic feasibility of the technology for industrial application. As an innovative approach in a demanding technological field, it results that the paper may deserves publication in the Microbiology Spectrum but after consideration of the 6 following comments.

C1 Introduction (line 64-68)

This is the part where the need for disinfection of fresh produce should be most emphasized to contribute to food safety. It is recommended to present the current status of food poisoning caused by fresh produce using the latest statistics.

C2 Materials and methods (Figure 1, line 145-147)

Unlike conventional PAWs, this device has a structure in which plasma is injected into water in real time. Therefore, the section where electrical discharge takes place is located very closely with water. Could this structure be a dangerous environment for workers?

C3 Materials and methods (line 181-184)

It is recommended to make the expression clearer.

[ten × 10 mL]: Is 100 mL of bacterial suspension (200 mL for *L. monocytogenes*) used in total?

C4 Materials and methods (Figure 1, line 188-189)

It is recommended to specify the position of the sample in the solution. In particular, the author explains that depending on the position of the sample in the solution, the electric field may be affected differently. In addition, the behavior of the sample may vary as bubbles enter the solution. As a result, the bactericidal effect of this device may be reduced in environments where the sample is located at the top of the solution. This means that the effect may vary depending on how many layers the sample is stacked in the solution. In addition, the effect may vary if the sample is light and floats in water.

C5 Results / Discussion (Figure 6, line 622-624)

The author explained that the amount of ozone dissolved in PAW is small due to the low solubility of ozone. Ozone is the most significant gas generated by a typical DBD system, and it is thought that a significant amount of ozone will also be generated in this system. Therefore, much of the ozone that is eventually insoluble in water can be exposed above the liquid. Please give us your opinion on this part and suggest alternative measures to the possibility of ozone exposure as the author considers industrial applicability important.

C6 Discussion (Figure 8, line 596-603)

Is there any way to directly observe how the electric field actually affected microbial cells? If the electric field is forming to a degree that has a profound effect on cells as the author suggests, I wonder if some cells should be killed even in an environment with tiron ($\cdot\text{O}_2^-$ scavenger) added. Perhaps this is most similar to the environment in which only electrical discharge, other than reactive species, affects. How about observing cell membrane integrity after electrical discharge with the addition of tiron?

End of report.

Response to reviewers (Rothwell et al, Spectrum00034-23)

We thank both reviewers for their detailed and helpful comments, which we address below. We believe these changes have greatly improved our manuscript and hope that it is now acceptable for publication in mSpectrum.

Reviewer #4:

The submitted manuscript reports on the use of dielectric barrier discharge diffusor for decontamination of bacteria in solution and on the cucamelon surface. The author tried to design experiments considering the economic feasibility of the technology for industrial application. As an innovative approach in a demanding technological field, it results that the paper may deserves publication in the Microbiology Spectrum but after consideration of the 6 following comments.

C1 Introduction (line 64-68)

This is the part where the need for disinfection of fresh produce should be most emphasized to contribute to food safety. It is recommended to present the current status of food poisoning caused by fresh produce using the latest statistics.

- The first paragraph of the introduction has been reworked to better emphasise the importance of sanitizers for fresh produce safety and we have added statistics on foodborne disease outbreaks due to fresh produce (lines 64-70). Unfortunately, we were not able to find a more recent paper that specifically reported on fresh produce than Li et al 2018.

C2 Materials and methods (Figure 1, line 145-147)

Unlike conventional PAWs, this device has a structure in which plasma is injected into water in real time. Therefore, the section where electrical discharge takes place is located very closely with water. Could this structure be a dangerous environment for workers?

- We agree that this will be an important consideration for future studies involving scale up and application of the technology and have added a sentence at the end of the paper (line 591-592) describing this as a possible limitation, along with the build-up of ozone mentioned below in C5.

C3 Materials and methods (line 181-184)

It is recommended to make the expression clearer.

[ten × 10 mL]: Is 100 mL of bacterial suspension (200 mL for *L. monocytogenes*) used in total?

- The text has been amended as follows: "Each cucamelon was then spot-inoculated with ten × 10 μL (100 μL total) of the *E. coli* or *S. enterica* inoculum at a final concentration of 1×10^9 colony forming units (CFU) mL⁻¹. Cucamelons were spot-inoculated with twenty × 10 μL spots of (200 μL total) of *L. monocytogenes* at 8×10^8 CFU mL⁻¹, as this species has a lower adhesion to cucumber (33)." (Lines 180-184).

C4 Materials and methods (Figure 1, line 188-189)

It is recommended to specify the position of the sample in the solution. In particular, the author explains that depending on the position of the sample in the solution, the electric field may be affected differently. In addition, the behavior of the sample may vary as bubbles enter the solution. As a result, the bactericidal effect of this device may be reduced in environments where the sample is located at the top of the solution. This means that the effect may vary depending on how many layers the sample is stacked in the solution. In

addition, the effect may vary if the sample is light and floats in water.

- We agree with the reviewer comment that the position of the sample in relation to the plasma electrodes is an important consideration in this study and for *in situ* PAW studies in general. Movement of the cucamelons in the wash solution is dynamic due to the bubbling of the plasma reactor. In our modelling study we addressed this by examining two cucamelon positions, one with the cucamelon next to the reactor (Figure 7) and one with it towards the top of the water (Figure S3). The bacterial cells positioned on the cucamelon near the water surface would be expected to experience the lowest levels of electrical forces in the system, which we calculated to be approximately 0.55 kV/cm, which is around a quarter of the strength of the maximum electric field experienced when cucamelon was next to the reactor (2 kV/cm). This is explained in the text from lines 466-470: "These demonstrate that a high local electric field of over 2.0 kV/cm can be formed inside the bacterial cell under these conditions. This is strongly dependent on the relative position of individual bacterial cells; when the cucamelon is located at the top of the water the maximum local electric field experienced by the bacteria is much lower at 0.24 to 0.55 kV/cm". We have modified the discussion to specify that "For future scale-up of this technology, reactor design and the position of the electrodes in the wash systems must be considered to guarantee effective antimicrobial power and thereby maximize food safety" (lines 593-5). We believe this is a sufficient caveat for the scope of this study.

C5 Results / Discussion (Figure 6, line 622-624)

The author explained that the amount of ozone dissolved in PAW is small due to the low solubility of ozone. Ozone is the most significant gas generated by a typical DBD system, and it is thought that a significant amount of ozone will also be generated in this system. Therefore, much of the ozone that is eventually insoluble in water can be exposed above the liquid. Please give us your opinion on this part and suggest alternative measures to the possibility of ozone exposure as the author considers industrial applicability important.

- We agree with the reviewer that the production of ozone and other potentially toxic gasses from the PAW systems will be an important safety consideration for upscale and applications of the technologies in future studies, and ventilation and fume extraction systems will likely need to be employed. We have noted this safety consideration alongside considerations of electrical hazards at the end of the discussion (lines 591-593).

C6 Discussion (Figure 8, line 596-603)

Is there any way to directly observe how the electric field actually affected microbial cells? If the electric field is forming to a degree that has a profound effect on cells as the author suggests, I wonder if some cells should be killed even in an environment with tiron ($\cdot\text{O}_2^-$ scavenger) added. Perhaps this is most similar to the environment in which only electrical discharge, other than reactive species, affects. How about observing cell membrane integrity after electrical discharge with the addition of tiron?

- The effects of the pulsed electric field on cells have previously been explored by another study (Mentheour R, Machala Z. Coupled Antibacterial Effects of Plasma-Activated Water and Pulsed Electric Field. *Frontiers in Physics*. 2022;10.) which was mentioned in our study on lines 131-133. While we agree with the reviewer that this would be an interesting area of future research, we believe that this is outside of the scope of the current study.

Reviewer #5:

The manuscript by Rothwell et al. describes multidisciplinary research in the field of food science evaluating

the use of plasma activated water for sanitizing cucamelons. It is the perception of this reviewer that the modelling components described in the current version would typically not be assessed or used by microbiologists. The modelling could have better served as an initial approach to optimize treatments so that a greater log reduction of pathogens was achievable. As presented the modeling is used to demonstrate mechanism. The ability of an audience of microbiologists to benefit from the modeling data is questionable.

- We understand the concern regarding the applicability of the modelling components described in our study to microbiologists. However, we believe that the modelling approach has significant value for both microbiologists and PAW researchers. Optimisation of PAW parameters to achieve the highest antimicrobial activity was completed in our previous study (Rothwell et al. "The antimicrobial efficacy of plasma-activated water against *Listeria* and *E. coli* is modulated by reactor design and water composition." *Journal of Applied Microbiology* 132.4 (2022): 2490-2500). The modelling in the current study built on this and on our new SEM work to better understand how the highly effective antimicrobial capacity of this system was generated and to advance knowledge in this emerging and multidisciplinary field. While plasma modelling is not a typical component of a microbiology research paper, we feel that presenting a comprehensive study that combines experimental data with modelling can provide a deeper understanding of the mechanisms at play, and that this approach can facilitate cross-disciplinary collaboration and spark new research questions that can drive innovation in the field.

The description of the modeling activities is vague in the materials and methods section for a microbiology audience. It is unclear how the discharge volume was estimated (line 309), how the residence time was calculated (line 310), and how were the settings applied for the activated plasma were chosen.

- To address the discharge volume and residence time comment, additional information has been added to the text:
"The discharge volume was estimated to be 1.1 cm³ based on the reactor's dimensions, which included an outer radius of 0.4 cm, an inner radius of 0.3 cm, and a height of 5 cm. By dividing the discharge volume by the gas flow rate, the residence time of gas species within the discharge volume was determined to be 0.067 seconds at the gas flow rate of 1 SLM" (Lines 306-310).
- To address the comment on how the settings for the plasma activation were chosen, further clarifications have been made in the text as follows:
"PAW was generated with a DBDD probe (PlasmaLeap Technologies) and power was supplied from a Leap100 micropulse generator (PlasmaLeap Technologies). The power supply settings used were based on the findings of our previous study (30). These settings included a frequency of 1500 Hz, a voltage of 120 volts, a duty cycle of 100 microseconds, and a resonance frequency of 60 kHz. Compressed air at a flow rate of 1 standard litre per minute (SLM) was used as the processing gas." (Lines 148-153).

Also, how is the E/N ratio used?

- The E/N ratio is an important parameter used in all of the plasma modelling in the current study. It represents the ratio of the electric field strength (E) to the gas density (N) in the plasma. The E/N ratio provides insights into the energy levels and behaviour of electrons, making it a crucial factor in plasma modelling.
- The following text has been amended in the methods to better explain how the E/N ratio was used:
"The reduced electric field (E/N) is calculated by dividing the electric field strength (E) by the density of neutral gas molecules (N) in the plasma. The E/N ratio is related to the energy distribution of electrons in the plasma, and it is an important parameter for plasma modelling. However, plasma exhibits highly transient and non-uniform characteristics, as depicted in Figure S1. Therefore, an averaged E/N value that accounts for these spatial and temporal variations was estimated using the QV Lissajous plot technique (39). The estimated E/N was approximately 30 Td as shown in Table S1

and so E/N values of 30, 40 and 50 Townsends (Td) were used for the initial plasma chemistry simulation. The influence of these parameters on the production of ·O₂- and other important gas products listed in Table 2 was investigated. As the gas composition at 30 Td was closest to that determined empirically in our previous study (30) we used 30 Td as the assumed E/N value used for subsequent plasma simulations.” (Line 314-325)

- The following additional table has been included in the supplementary figures to help address this reviewer comment:

Table S1. Estimated characteristic capacitance of the given plasma system and calculated reduced electric field (E/N) following Wagner *et al.* (39) where 1 [Td] = 1×10^{-17} Vcm²

	U _{min} [kV]	C _o [pF]	C _p [pF]	U _{disch} [kV]	Gap[cm]	E[V/cm]	E/N[Td]
max. current period	1.25	23.5	37.3	0.767	0.1	7668.6	31.3
max. voltage period	1.09	22.6	36.0	0.700	0.1	6696.2	27.4

A number of gas products are considered in Table S1 but the text implies that only one of those is relevant in the system used. Please resolve this incongruence, is table S1 needed? Are nitrogen oxides produced in the plasma system chosen, possibly from N₂ in air?

- The species listed in what is now Table S2 are the total species included in the model. In Fig 6(A), we have chosen only major gas species from the model including NO_x in order to make a clear comparison without complexity, where NO_x indicates the sum of total nitrogen oxides including NO, NO₂, NO₃, N₂O₃, N₂O₄ and N₂O₅. In order to avoid confusion, the figure caption is now corrected as: “Figure 6. Plasma chemistry modelling of the N₂/O₂ plasma discharge at atmospheric pressure. (A) The mole fraction of important gas species produced by the DBDD plasma at E/N conditions of 30, 40 and 50 Td. NO_x indicates total nitrogen oxide species in the gas phase including NO, NO₂, NO₃, N₂O₃, N₂O₄ and N₂O₅.” (Line 856)
- All of the gas reactions listed in Table S2 are from reactions in the N₂/O₂ (air) plasma system, therefore the nitrogen oxides are from the reaction of the air molecules in the plasma discharge.

It seems that the rapid solvation of the oxygen species (line 629) works against the effectiveness of the treatment as the log reduction achieved is minimal and the produce spoils in about 10 days. Can the PAW be optimized to increase the shelf-life of the produce?

- We respectfully disagree with the comment that log reduction is minimal and the produce spoils in about 10 days. As demonstrated in Figure 3, PAW treatment was highly effective, reducing the counts of pathogenic bacteria on the cucamelon surface by 3 log CFU g⁻¹, and completely eliminating pathogens from the wash water. It demonstrated comparable antimicrobial efficacy to that of the established sanitiser NaOCl, while using a rapid 2-minute wash time. With regard to shelf-life, as noted in the methods a quality scores of 3 and lower indicates that the product is no longer within acceptable specifications for consumption, and Figure 5 E demonstrates that the quality of the cucamelons treated with PAW remained above 3 throughout the 28 day storage trial. We have modified the text from lines 413-417 to make this clearer.
- It is possible that PAW could be further optimised but there will always be a trade-off between antimicrobial efficacy and the quality of delicate fresh produce.

The modeling section should occur first in the materials and methods, so that it aligns with Figure 2 as an introductory piece.

- The structure of the manuscript begins with the antimicrobial capacity of PAW and its effect on produce quality, which is followed by a more in-depth analysis on the mode of action through modelling. We feel this order is most appropriate for a microbiology journal. Please also note that

modelling is itself a result and was developed based on the results of the SEM and antimicrobial work in order to better understand how PAW affects bacterial cells.

Although the protocols used for microbial counts, and texture, color and appearance evaluation may be standard, references are needed to support them in the materials and methods section.

- As cucamelons are not a commonly studied fresh produce item, the evaluation methods for texture, color and appearance were developed by our group for this study, and no prior references exist.

A sentence should be added in line 132 about the use of tiron as a control treatment. It seems to come out of nowhere when the reader first encounters it in line 201. Tiron inclusion is working as a control treatment and should be presented as such.

- To address this comment, additional text has been added to the introduction as follows: "... assays using the ·O₂- scavenger tiron demonstrated that ·O₂- was essential for the antimicrobial activity of this system." (Line 121)

Cucamelons, like other fruits and vegetables, are susceptible to equilibration with water in its surroundings (reference 32). Such dynamic needs to be considered regarding the use of solutions that were refrigerated (4C) prior to treatment application (line 170). The equilibration of fruits and vegetables with water depends on temperature. Was the temperature of the fruits during treatment measured? Were the solutions tempered?

- In our methods we specify that the "All treatments were made using autoclaved tap water cooled to 4 °C in a final volume of 200 mL" (line 143). We have clarified on line 177 that the cucamelons "were stored at 9 °C with 85% humidity" prior to the experiment.

Also, was there any indication in the electron microscopy work that the pathogens had penetrated the cucamelons' flesh? The effectiveness of the sanitizer to that of the plasma treatment may not be comparable as one treatment can reduce the microbes in the 1mm outer flesh near the skin, but plasma cannot. Please elaborate on this aspect. (Mattos FR, Fasina OO, Reina LD, Fleming HP, Breidt F Jr, Damasceno GS, Passos FV. 2005. Heat transfer and microbial kinetics modeling to determine the location of microorganisms within cucumber fruit. J Food Sci 70(5):E324-E330.)

- As specified in the introduction of the article, ("Sanitizers are critical for reducing the risk of cross contamination by pathogens that may have been transferred into the wash solution", Line 72) the critical role of sanitisers is to prevent cross-contamination by killing pathogens in the wash water. In this study, we demonstrate the capacity for PAW to reduce pathogens to below detectable limits in the wash water using a fresh produce model. While assessment of the penetration of pathogens into the cucamelon surface is an interesting concept for future studies, we believe that this is outside the scope of the current paper where the focus was to determine the efficacy of PAW with reference to the established sanitiser NaOCl.

The use of tap water (line 103 &139) and lactic acid (line 145): Was the mineral content of the tap water used evaluated? Would researchers be able to reproduce your findings with any tap water? Lactic acid (10% or 1.1 M) is an antimicrobial that was added to the chlorine solution for pH control. Thus, the sanitizer treatment could have been more effective than it typically is. Is lactic acid routinely used by processors in the wash waters for pH control?

- To address the comment on the tap water used in this study, additional information has been added to line 143-144 of the text: "The chemical analysis of the Sydney tap water used in this study has been published previously (30)."

- Producers must use food safe acid to acidify their sanitiser solutions in production, therefore the use of lactic acid was deemed appropriate in this study. A small number of drops of the 10% lactic acid (<1mL) were added to the solution to reduce the pH, which is insignificant compared to the concentrated chlorine solution that provides the antimicrobial action. Acidification of chlorine solutions is standard practice in food safety sanitiser work.

Is there a reason for not studying the flavor of the treated and control samples?

- The plasma machine has been set up in a PC2 laboratory and all experiments were conducted in this laboratory. It is strictly forbidden to consume any materials that have been produced in the PC2 environment.

Minor comments:

Line 190: How was the probe sanitized/sterilized between treatments?

- Additional information has been added to the text to address this comment: "The probe was sanitized between treatments by wiping with ethanol and then rinsing with sterile tap water." (Line 191-192)

Line 206: What was the definition of biological controls?

- We assume that this is referring to the use of the term "biological triplicates". Biological replicates are generally defined as a repeat of the entire experiment on a separate day. We agree that the term "biological triplicates" is confusing and the text has been modified as such: "All experiments were performed in duplicate, with three biological replicates performed on separate days." (Line 207)

Line 231, 239 and throughout the manuscript: Please use the term microbiota instead of microflora (small plants).

- "Microflora" has been changed to "microbiota" as per the reviewer's recommendation on line 231, 392, 399 and 482.

Line 239: Assessment of the microbial load on cucamelons as a function of treatment.

- The subheading "Background microflora assessment" has been changed to "Assessment of the microbial load on cucamelons as a function of treatment" as per the reviewer's recommendation. (Line 239)

Line 79: ...from food sanitation are hazardous...

- "...food sanitisation are also..." has been corrected to "... food sanitization are also..." (Line 81)

Line 296 to 302: panelist

- "panellist" has been changed to "panelist" (Line 295)

Line 423 to 425: Please show data for microbial counts from wash waters.

- As this result is very minor, the data were placed into the text and was not included as a graph. On line 402, we have included "data not shown" to address the reviewer's comment.

Line 662: Please spell out CTO

- CTO has been replaced with Chief Technology Officer. (line 610)

June 18, 2023

Prof. Dee A Carter
The University of Sydney
School of Life And Environmental Sciences
LEES Building F22, School of Life And Environmental Sciences
University of Sydney
Sydney, New South Wales 2006
Australia

Re: Spectrum00034-23R1 (An effective sanitizer for fresh produce production: In situ plasma activated water treatment inactivates pathogenic bacteria and maintains the quality of cucurbit fruit)

Dear Prof. Dee A Carter:

Your manuscript has been accepted, and I am forwarding it to the ASM Journals Department for publication. You will be notified when your proofs are ready to be viewed.

Sincerely,

Jasna Kovac
Editor, Microbiology Spectrum
